# Video Polyp Segmentation using Implicit Networks

**Aviad Dahan**                                         AVIADDAHAN@MAIL.TAU.AC.IL
**Tal Shaharabany**                                   SHAHARABANY@MAIL.TAU.AC.IL
**Raja Giryes**                                             RAJA@TAUEX.TAU.AC.IL
**Lior Wolf**                                                 WOLF@CS.TAU.AC.IL
*Tel Aviv University, Israel*

**Editors:** Accepted for publication at MIDL 2024

## Abstract

Polyp segmentation in endoscopic videos is an essential task in medical image and video analysis, requiring pixel-level accuracy to accurately identify and localize polyps within the video sequences. Addressing this task unveils the intricate interplay of dynamic changes in the video and the complexities involved in tracking polyps across frames. Our research presents an innovative approach to effectively meet these challenges that integrates, at test time, a pre-trained image (2D) model with a new form of implicit representation. By leveraging the temporal understanding provided by implicit networks and enhancing it with optical flow-based temporal losses, we significantly enhance the precision and consistency of polyp segmentation across sequential frames. Our proposed framework demonstrates excellent performance across various medical benchmarks and datasets, setting a new standard in video polyp segmentation with high spatial and temporal consistency. Our code is publicly available at https://github.com/AviadDahan/VPS-implicit

**Keywords:** Polyp segmentation, Video Segmentation, Implicit Networks, Optical Flow

## 1. Introduction

The segmentation of polyps in videos plays an important challenge in medical imaging, aiming to precisely identify and localize polyps with pixel-level accuracy within video sequences. This task is crucial for applications such as endoscopic surgery planning. Addressing it involves tackling the complexity of tracking these unique biological structures through various frames, often amidst partial occlusions or changes in viewpoint.

The primary difficulty in polyp segmentation lies in the diverse transformations that polyps undergo, such as changes in size, rotation, occlusion, and lighting conditions within the gastrointestinal tract. These variations challenge the consistency of segmentation predictions, particularly in endoscopic videos where the camera's movement may cause significant changes in the background and foreground, potentially leading to inaccurate segmentation.

In this paper, we introduce a novel framework for video polyp segmentation, harnessing the capabilities of implicit networks to enhance temporal coherence and segmentation precision throughout video sequences. Building on a 2D pre-trained network, referred to as $g$, our approach integrates an implicit network, $h$, augmented with optical flow-based temporal loss functions. This integration is done at test-time, by optimizing network $h$ and without training $g$ or making additional inferences with it.

At the heart of our innovation is the effective integration of 2D and implicit networks, which incorporates temporal information into the segmentation process, thereby enhancing

the accuracy and consistency of polyp detection over time. By applying optical flow-based losses, our framework ensures more reliable tracking of polyps across frames, addressing the challenges of temporal coherence with greater efficiency.

While the framework is general and can be easily adapted to different segmentation tasks and base segmentation networks, we focus on the task of video polyp segmentation using a pretrained image-based AutoSAM (Shaharabany et al., 2023) network or NanoNet (Jha et al., 2021). The polyp segmentation task provides a well-researched medical imaging domain, with well-established benchmarks, and, in the case of AutoSAM, the opportunity to apply a powerful foundation model derived from the Segment Anything (SAM) network (Kirillov et al., 2023), for which this domain is out of distribution.

We demonstrate the effectiveness of our methodology through extensive testing on various datasets of video polyp segmentation tasks, including capsule endoscopy and colonoscopy. Despite its conceptual simplicity, our approach sets new benchmarks in these areas, proving its potential and efficiency in advancing video polyp segmentation.

## 2. Related Work

**Optical Flow and Video Segmentation** The use of optical flow to enhance video segmentation has been explored in the past and we discuss three representative examples. Tsai et al. (2016) note that while optical flow can be used to propagate segmentation masks, it can be inaccurate, so they propose to refine the optical flow by using segmentation boundaries. Unlike our work, they use a Conditional Random Field (CRF) for the spatio-temporal segmentation model.

Cheng et al. (2017) consider a different mechanism, in which information is shared between a segmentation network and a two-frame optical flow network. Integration occurs through connections between the upsampling layers of the two networks, which are independent up to this point and train with different loss terms. In our method, optical flow is not computed; instead, it is used to add a spatial constraint to the segmentation maps. This constraint is expressed as a loss term, not by using network activations.

Another strategy to improve object delineation using optical flow is presented by Fan et al. (2021) who use pre-computed optical flow to create an integrated image that combines multiple frames. They present evidence that performing video object-detection on the resulting image boosts detection accuracy.

**Recent Advancements in Image Segmentation** In some of the experiments, our work employs the AutoSAM architecture introduced by Shaharabany et al. (2023), a modification of the Segment Anything (SAM) network (Kirillov et al., 2023), which replaces SAM's prompt encoder with an initial segmentation network.

SAM (Kirillov et al., 2023), similarly to other contributions (Strudel et al., 2021), employs a visual-transformer (Strudel et al., 2021). It is trained on a massive supervised segmentation dataset, which contains over 1 billion masks for objects in 11 million natural images. The model quickly became an effective foundation model for other segmentation solutions and downstream tasks (Wu et al., 2023; Liu et al., 2023; Huang et al., 2023).

**Polyp Segmentation** The task of polyp segmentation involves the precise delineation of polyps from surrounding tissues in images or video frames captured during endoscopic

examinations. The work by Jha et al. (2019), ResUNet++, which builds on the ResUNet architecture, showcases superior performance in pixel-wise polyp segmentation across standard datasets. Following this, the authors introduced NanoNet (Jha et al., 2021), a novel, lightweight architecture designed for real-time segmentation of capsule endoscopy and colonoscopy. In the realm of video polyp segmentation, Ji et al. (2021a) present a pioneering study, introducing the SUN-SEG dataset, as well as a new baseline model, PNS+. Shaharabany et al. (2023) also demonstrate remarkable results in the field of polyp segmentation and video polyp segmentation, despite being image-based.

**Neural Implicit Representations** Representing a function by a neural network has been found useful in various applications. A prominent example is neural radiance fields (NeRF) (Mildenhall et al., 2020), which depict a 3D scene via a neural-based function (Wang et al., 2022; Mildenhall et al., 2022; Gu et al., 2021; Park et al., 2021; Barron et al., 2022; Verbin et al., 2022; Tewari et al., 2022).

In the realm of 2D, neural textures based on rasterization serve a similar purpose, facilitating the generation of lifelike images (Thies et al., 2019; Rivoir et al., 2021). With an emphasis on representing surface textures through continuous parametric functions without pixel constraints, Burgert et al. (2022) present neural-neural textures, leveraging the potential of Fourier features (Tancik et al., 2020). Burgert et al. (2023) show the potential of representing a segmentation mask using a combination of Fourier features and implicit models. Building on this foundation and incorporating temporal data in the form of optical flow, our model effectively captures consistent alpha-channels, which play a pivotal role in our step-by-step segmentation scheme.

## 3. Method

Our proposed approach for video polyp segmentation builds upon a 2D object extraction network $g$, e.g., a segmentation network, and incorporates an implicit network $h$ that is trained to predict the output of $g$ for all frames, but with the addition of optical flow losses that ensure temporal consistency between consecutive frames. As the network $g$ one can employ the relevant SOTA algorithm for the associated 2D task. Next, we discuss the training of $h$ for a given pre-trained $g$.

**Test-time training of implicit function $h$.** For each frame $J \in \mathbb{R}^{W \times H \times 3}$ the segmentation network $g$ predicts a mask $M \in \mathbb{R}^{W \times H}$ indicating the pixels that depict the object of interest inside the frame, where $W$ and $H$ denote the width and height of each frame, respectively. Notably, these masks are independent and lack temporal coherence across successive frames, leading to suboptimal accuracy.

The implicit network $h : x, y, t \to \alpha$ is trained for each input RGB video with $N$ frames $V \in \mathbb{R}^{N \times W \times H \times 3}$. $h$ aims to refine these frame-wise predictions by introducing temporal information via optical flow. $h$ receives coordinates $(x, y, t)$ as input and returns the probability $\alpha$ of the coordinate belonging to either the foreground or background.

The training of $h$ is done during test time, running in an unsupervised manner on the desired test video. During the training of the implicit network $h$, a composite loss function is employed, comprised of a combination of a labeling loss that compares the output of $h$ to that of $g$ and is based on the Binary Cross-Entropy (BCE) loss, and an optical flow loss.

Figure 1: An illustration of $h$. The coordinates are positionally encoded, resulting in Fourier features. These pass through an 8-layer MLP that introduces a mask probability for each coordinate.

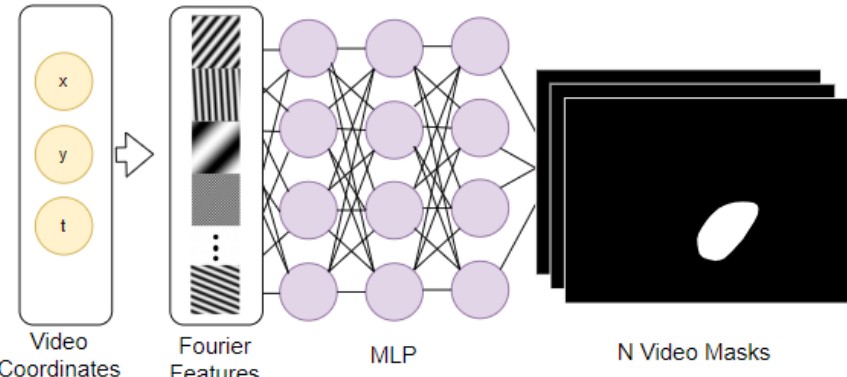

**Loss terms.** The overall loss function can be expressed as:

$$\mathcal{L} = \mathcal{L}_{label} + \lambda * \mathcal{L}_{flow}. \tag{1}$$

The first loss $\mathcal{L}_{label}$ aligns the output of $h$ with the prediction of $g$. Simultaneously, the second loss $\mathcal{L}_{flow}$ with the coefficient $\lambda$ enforces temporal consistency across frames. Neither $g$ nor the optical flow extractor go through any further training at that stage, i.e., the optical flow data and $g$'s output mask are being used as is. The first term in (1), the label loss, is defined as

$$\mathcal{L}_{label}(h, g, V) = \sum_{(x,y,t)} BCE(h(x,y,t), M^t(x,y)), \tag{2}$$

where $(x, y, t)$ run over all spatial and temporal locations, and $M^t = g(V(t, \cdot, \cdot, \cdot))$ is the segmentation mask that is obtained by applying $g$ to frame $t$ of the video $V$.

We calculate the optical flow mapping $\Delta X^t, \Delta Y^t \in \mathbb{R}^{W \times H}$ between every frame $t$ and its next frame $t+1$ and use it to warp the predicted segmentation mask of one frame to the next frame. We then compute the mean squared error between the warped segmentation mask of a frame and that of the next frame. This can be written as:

$$\mathcal{L}_{flow}(h, \{(\Delta X^t, \Delta Y^t)\}_t) = \tag{3}$$

$$\frac{1}{NWH} \sum_{(x,y,t)} \|h(x,y,t) - h(x + \Delta X^t, y + \Delta Y^t, t+1)\|,$$

where, as in (2), $(x, y, t)$ cover the entire range $[1..N] \times [1..W] \times [1..H]$.

Our solution presents a significant advantage in its versatility, as it is not dependent on the specific task that the 2D object extraction network $g$ is addressing. The same procedure described above may be applied to different tasks, where each has its own $g$ function that might be significantly different from the other ones.

**Fourier Features.** To enhance the network's leaning capability, following Burgert et al. (2023), we have incorporated a Fourier feature transformation to the input. Each coordinate (x,y,t) is normalized in the range [-1,1] and then converted to a vector of N=16 Fourier

Figure 2: The results of our implicit network $h$ on consecutive frames taken from the SUN-SEG Hard dataset. The 1st row shows the input image $I$, the 2nd row shows the ground truth segmentation masks $M$, the 3rd depicts the results of the image-based model $g(I)$, and the last row shows the output of our implicit network $h$. Evidently, the inconsistency occurring in frame N is eliminated by our method.

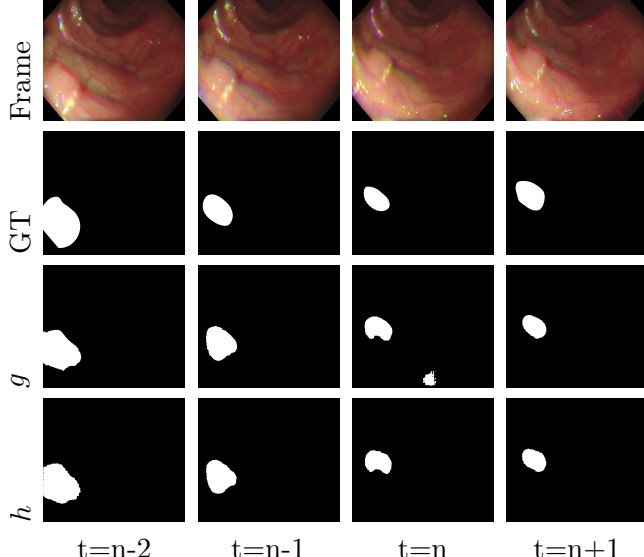

features represented by an exponential distribution between a base frequency ($n = 0$) and the $2^N$ harmony. This scale allows us to emphasize the representation of the lower frequencies where most of the data is present. For example, for the x coordinate, each frequency is represented by a pair $\{\sin(2^n\pi x), \cos(2^n\pi x)\}$ where $n \in \{0, 1, ..., N-1\}$.

## 4. Experiments

In this study, we evaluate our proposed method on multiple video datasets. We compare our results with state-of-the-art methods and also present a number of exploratory results in video polyp segmentation.

**Datasets**  Our method was tested on three datasets:

**SUN-SEG**  The first two datasets are SUN-SEG-Easy (unseen) and SUN-SEG-Hard (unseen), both from the SUN-SEG Video-Polyp-Segmentation database (Misawa et al., 2021; Ji et al., 2022), which contains $1,106$ short video clips, with $158,690$ video frames in total. We used the official train-test split. We compare our results with both image-based methods, namely, UNet (Ronneberger et al., 2015), UNet++ (Zhou et al., 2019), ACSNet (Zhang et al., 2020), PraNet (Fan et al., 2020), SANet (Wei et al., 2021) and AutoSAM (Shaharabany et al., 2023) as well as video-based methods, including COSNet (Lu et al., 2019), MAT (Zhou et al., 2020), PCSA (Gu et al., 2020), 2/3D (Puyal et al., 2020), AMD (Liu et al., 2021), DCF (Zhang et al., 2021), FSNet (Ji et al., 2021b) and PNSNet (Ji et al., 2021a).

Table 1: Quantitative results of two test sub-datasets from the SUN-SEG (Ji et al., 2022) dataset. Our method shows improvements over prior methods, achieving SOTA results in almost every metric. The best values are highlighted in **bold**.

| | | SUN-SEG-Easy | | | | | | SUN-SEG-Hard | | | | | |
|---|---|---|---|---|---|---|---|---|---|---|---|---|---|
| | | $\mathcal{S}_{\lambda_f}$ | $E_\phi^{mn}$ | $F_\beta^w$ | $F_\beta^{mn}$ | Dice | Sen | $\mathcal{S}_{\lambda_f}$ | $E_\phi^{mn}$ | $F_\beta^w$ | $F_\beta^{mn}$ | Dice | Sen |
| Image-based | UNet | 0.67 | 0.68 | 0.46 | 0.53 | 0.53 | 0.42 | 0.67 | 0.68 | 0.46 | 0.53 | 0.54 | 0.43 |
| | UNet++ | 0.68 | 0.69 | 0.49 | 0.55 | 0.56 | 0.46 | 0.69 | 0.70 | 0.48 | 0.54 | 0.55 | 0.47 |
| | ACSNet | 0.78 | 0.78 | 0.64 | 0.69 | 0.71 | 0.60 | 0.78 | 0.79 | 0.64 | 0.68 | 0.71 | 0.62 |
| | PraNet | 0.73 | 0.75 | 0.57 | 0.63 | 0.62 | 0.52 | 0.72 | 0.73 | 0.54 | 0.61 | 0.60 | 0.51 |
| | SANet | 0.72 | 0.74 | 0.57 | 0.63 | 0.65 | 0.52 | 0.71 | 0.74 | 0.53 | 0.58 | 0.60 | 0.50 |
| | AutoSAM | **0.82** | **0.86** | **0.72** | **0.77** | 0.75 | 0.67 | 0.82 | **0.87** | 0.71 | 0.76 | 0.76 | **0.73** |
| Video-based | COSNet | 0.65 | 0.60 | 0.43 | 0.50 | 0.60 | 0.36 | 0.67 | 0.63 | 0.44 | 0.51 | 0.61 | 0.38 |
| | MAT | 0.77 | 0.74 | 0.57 | 0.64 | 0.71 | 0.54 | 0.78 | 0.75 | 0.58 | 0.64 | 0.71 | 0.58 |
| | PCSA | 0.68 | 0.66 | 0.45 | 0.52 | 0.59 | 0.40 | 0.68 | 0.66 | 0.44 | 0.51 | 0.58 | 0.41 |
| | 2/3D | 0.78 | 0.78 | 0.65 | 0.71 | 0.72 | 0.60 | 0.79 | 0.77 | 0.63 | 0.69 | 0.71 | 0.61 |
| | AMD | 0.47 | 0.53 | 0.13 | 0.15 | 0.27 | 0.22 | 0.47 | 0.527 | 0.13 | 0.14 | 0.25 | 0.21 |
| | DCF | 0.52 | 0.51 | 0.27 | 0.31 | 0.32 | 0.34 | 0.51 | 0.52 | 0.26 | 0.30 | 0.32 | 0.36 |
| | FSNet | 0.72 | 0.70 | 0.55 | 0.63 | 0.70 | 0.49 | 0.72 | 0.69 | 0.54 | 0.61 | 0.70 | 0.49 |
| | PNSNet | 0.77 | 0.74 | 0.62 | 0.66 | 0.68 | 0.57 | 0.77 | 0.75 | 0.61 | 0.66 | 0.67 | 0.58 |
| | VPS+ | 0.81 | 0.80 | 0.68 | 0.73 | 0.76 | 0.63 | 0.78 | 0.79 | 0.65 | 0.71 | 0.74 | 0.62 |
| | Ours | **0.82** | 0.85 | 0.71 | **0.77** | **0.77** | **0.68** | **0.84** | **0.87** | **0.73** | **0.78** | **0.79** | 0.72 |

**KvasirCapsule-SEG**    The third dataset is KvasirCapsule-SEG, an enchanced subset of Kvasir-Capsule (Smedsrud et al., 2021), which was introduced by Jha et al. (2021) and contains 55 pillcam endoscopy frames, their segmentation ground truth, and bounding box information. Here, we used a custom 80-20 train-test split, taking the last 20% frames from the dataset as test data, in order to retain the order of the frames. We compare our results to AutoSAM (Shaharabany et al., 2023) and three versions of NanoNet (Jha et al., 2021), namely A, B and C, where A is the largest architecture and C is the smallest.

**Implementation details**    The architecture of the implicit network $h$ is inspired by the work of Burgert et al. (2023), expanded to video by including the time dimension. It is described in Fig. 1. The input to the network consists of spatial coordinates $(x, y)$ and the temporal dimension $t$. The model $h(x, y, t)$ consists of a positional encoding layer with 16 positional embeddings for each of the three dimensions in the form of Fourier features, followed by an 8-layer MLP with a hidden dimension set to 400. This continuous representation ensures high-resolution predictions alongside smooth transitions, making the model capable of producing detailed saliency masks without the need for upscaling.

The optical flow data is extracted using RAFT (Teed and Deng, 2020).

**Training details**    During inference, $h$ is being trained on each test video separately, yielding a set of masks for this specific video for benchmarking. At each step in the training input coordinates $(x, y, t)$ are randomly selected from a uniform distribution in the range [-1,1]. We employ the ADAM optimizer with an initial learning rate of 0.001, and a weight

decay regularization parameter set to $1 \cdot 10^{-5}$. A batch size of 1 is utilized - and a set of $50k$ points is trained at each step. We conduct training on NVIDIA A6000 with 48GB GPU RAM. The optical flow loss weight, $\lambda_{of}$, is set to 1.

For the SUN-SEG dataset, we used AutoSAM (Shaharabany et al., 2023) as $g$, which generates the image-based masks without using temporal information. Once the base masks and the optical flow have been extracted, each video is refined individually using $h$. The network $h$ trains for 1000 steps. In each step, a subset of the $(x, y, t)$ coordinates is used.

For KvasirCapsule-SEG, in order to explore the capabilites of our method, we use NanoNet (Jha et al., 2021) alongside AutoSAM as $g$ and train $h$ for 500 steps.

Table 2: Quantitative results for the KvasirCapsule-SEG (Jha et al., 2021) dataset. Our method shows improvements on every metric, for each of the different sizes of NanoNet. The best values are highlighted in **bold**.

| Method | DSC | mIOU | Recall | Precision | F2 | Accuracy |
|---|---|---|---|---|---|---|
| NanoNet-A | 0.9294 | 0.8697 | 0.9567 | 0.9080 | 0.9450 | 0.9233 |
| NanoNet-A + h | **0.9356** | **0.8807** | **0.9592** | **0.9159** | **0.9492** | **0.9286** |
| NanoNet-B | 0.9258 | 0.8634 | 0.9507 | 0.9042 | 0.9403 | 0.9200 |
| NanoNet-B + h | **0.9368** | **0.8820** | **0.9574** | **0.9186** | **0.9489** | **0.9310** |
| NanoNet-C | 0.9224 | 0.8576 | 0.9326 | 0.9154 | 0.9281 | 0.9167 |
| NanoNet-C + h | **0.9265** | **0.8645** | **0.9369** | **0.9190** | **0.9324** | **0.9205** |
| AutoSAM | 0.9429 | 0.8931 | 0.9272 | 0.9616 | 0.9332 | 0.9368 |
| AutoSAM + h | **0.9435** | **0.8941** | **0.9282** | **0.9617** | **0.9340** | **0.9375** |

**Evaluation metrics**

For SUN-SEG, following (Ji et al., 2022), we use six different metrics for model evaluation between prediction $P_s$ and ground-truth $G_s$ at timestamp $s$. These metrics are as follows: (a) Dice coefficient (b) Pixel-wise sensitivity (c) F-measure (d) Weighted F-measure (e) Structure measure (f) Enhanced-alignment measure. For KvasirCapsule-SEG we follow Jha et al. (2021), using the six metrics: DSC, mIoU, Recall, Precision, F2, and Accuracy.

**Results**

Table 1 shows the results for the SUN-SEG dataset. For the SUN-SEG-Hard (unseen) dataset our method outperforms the state-of-the-art (Shaharabany et al., 2023) in the following metrics: $\mathcal{S}_{\lambda_f}$ , $F_\beta^w$ , $F_\beta^{mn}$ & Dice with a margin of 2, 2, 2, 3, respectively, with a negative impact of 1% in Sensitivity. For the SUN-SEG-Easy (unseen) dataset our method outperforms the state-of-the-art methods in Dice Score and Sensitivity, with a margin of 2 and 1, respectively, and matches them in $\mathcal{S}_{\lambda_f}$ & $F_\beta^{mn}$, with a negative impact of 1% in $E_\phi^{mn}$ & $F_\beta^w$. To our understanding, the minimal negative impact is a result of the intricate interplay between optical flow and grounding to the existing segmentation mask. We investigate this behaviour in the ablation study for this work. Fig. 2 illustrates the advantage of using the implicit network on SUN-SEG data. Note that applying per-frame object segmentation leads to inconsistent results, which are mitigated by adding the implicit network $h$.

Figure 3: Ablation study investigating different levels of optical flow blending on the KvasirCapsule-SEG (Jha et al., 2021) dataset using Nanonet-B as baseline. Base stands for the baseline network without h being applied to it.

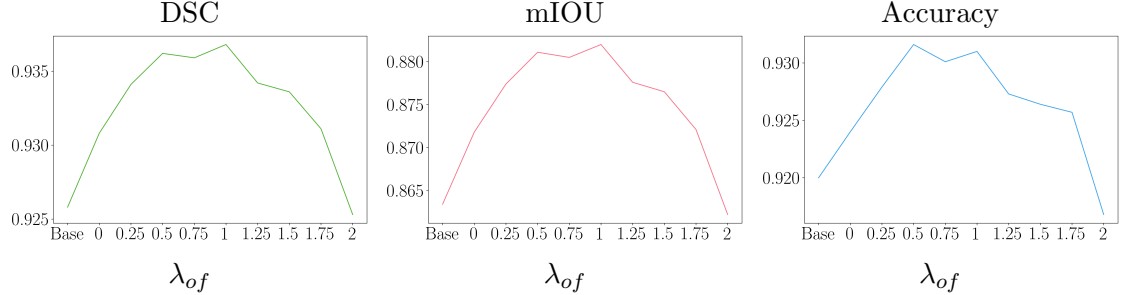

For the KvasirCapsule-SEG, Table 2 shows the results. In both cases, AutoSAM and NanoNet, our method improves the current methods on every metric. Notably, for NanoNet we demonstrate an improvement of 1-2% percent in mIOU and 0.5-1% in DSC and Accuracy.

**Ablation study**

In order to investigate the merits of the optical flow loss, we conducted a number of experiments, with varying levels of $\lambda_{of}$ while the rest of the parameters are set as mentioned in the training details section, as shown in Figure 3. Evidently, the introduction of optical flow loss improves performance. We also notice that increasing $\lambda_{of}$ too much has a negative impact on performance. In such a case the model is less grounded to the segmentation masks provided by $g$ and hence more prone to drifts in optical flow.

## 5. Limitations

While the use of $h$ improves the video performance of pre-trained 2D networks, the main limitation of our proposed approach is that the network $h$ is trained at test time. While efficient to train, this is a divergence from the feed-forward nature of other solutions, especially image-based ones. As future work, we may employ meta-learning to make $h$ feed forward, i.e., train, on the training set, a network that changes its behavior based on the input video, using, for example, hypernetworks (Ha et al., 2017).

## 6. Conclusions

We present an innovative approach for video polyp segmentation that lifts 2D segmentation networks to video solutions by employing implicit networks and enforcing temporal consistency through optical flow. Our experimental evaluations have demonstrated the effectiveness of the proposed method. The results consistently showcase improved performance on a variety of metrics compared to 2D segmentation-based approaches. Future work may investigate the expansion of our technique to 3D scans, such as CT or MRI, as well as multi-object segmentation, thereby further solidifying the applicability and impact of our proposed approach across different modalities and tasks in the medical field.

## Acknowledgments

Funded by the European Union (ERC, P3D Endoscopy, 101113391). Views and opinions expressed are however those of the author(s) only and do not necessarily reflect those of the European Union or the European Research Council Executive Agency. Neither the European Union nor the granting authority can be held responsible for them.

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
