# OpenReview forum: "Video Polyp Segmentation using Implicit Networks"
_MIDL.io/2024/Conference — MIDL 2024 Poster_

### Official Review · Reviewer_j1t2 · 2024-02-28

**Confidence:** 5
**Preliminary Rating:** 2
**Recommendation:** Poster

**Summary:**

Dahan et al., propose a pipeline for video segmentation that consists of a single 2D image segmentation and an additional unsupervised network to curate the results of the segmentation to preserve the temporal consistency across frames. The underlying idea is that the first network (g) could be pre-trained or a foundation model that computes a preliminary mask, and that the second one (h) can easily curate these segmentations without relying on annotated datasets.

**Strengths:**

- The method proposes a solution to perform video segmentation without relying on complex or large deep models. This has the benefit of being more flexible with training data, and getting lighter models that require less computing resources and are more environmentally friendly.
- The pipeline is highly generalisable and flexible in the sense that the additional network h for the temporal consistency can be connected to any other network trained for images from different domains.
- The authors compared the proposed method with quite an extended list of approaches.

**Weaknesses:**

- From the description of the second network, it is not clear how much the performance relies on the preliminary 2D segmentation. When looking at the results in Figure 2, it seems that the segmentations of g and h are quite similar and that the proposed improvement could have been achieved by removing the over-segmentation in t = n with simple connected components. Indeed, when looking at the results in Table1, this idea is reinforced as the results from AutoSAM are quite similar if not the same, and are actually the ones with the highest values.
- The dataset chosen does not show the potential of the method to address time consistency. Therefore, the contribution of the proposed approach is completely diluted.
- Details about the proposed h and how the training is performed should be clarified (see the detailed comments).
- Poor benchmark: details about the quantitative assessment should be clarified (see the detailed comments).

**Detailed Comments:**

- Input of the h-network: is it the raw values of the image at (x,y,t) or the x,y,t coordinates in the video? i.e. is it (x,y,t) or V(x,y,t)? This information is not clear in the text and in Figure 1 it seems it is the coordinates. If so, how are these coordinates normalised?
- How do the Fourier features relate to these coordinates? Are they local? I would suggest the authors to further explain these details in the text.
- Optical flow: the approach used to compute the optical flow was not trained for this domain, so how much does it contribute to improving the performance? Also, are the authors using the directed optical flow or just the magnitude of Delta X and Delta Y?
- h-training: it is meant to happen in inference with test images, then is it trained on each video or for the entire test set? Is there any data class balance for the selection of (x,y,t)?
- Benchmark:
   - Were the approaches chosen for comparison retrained for this specific data domain? If not, are any of those supposed to be generalisable such as the foundation model AutoSAM? If so, to provide a fair comparison, it would be recommendable to indicate which ones were retrained and which ones are meant to be generalisable across domains.
    - If possible, I would recommend the authors cite each of the approaches used for comparison.
    - The table only contains mean values and many of them are quite similar, which impedes any conclusion regarding the performance of the method. A solution is to provide plots showing the distribution of these accuracies or the standard deviations of each of these metrics.
    - Were the accuracy metrics for the image-based and video-based approaches computed in the same way? That is, where the accuracy metrics compute for each video and then averaged, or were all the metrics averaged for all the frames?

**Justification Of The Preliminary Rating:**

The authors suggested a sophisticated and combined approach to perform video segmentation, while given the results and the figures, such a method or equivalent complex approaches are not justified. Just using AutoSAM, without any further training, the task could be performed obtaining equivalent results.

**Questions To Address In The Rebuttal:**

All the questions have been included in the previous section

**Special Issue:**

No

---

> ### Author Response · Authors · 2024-03-10
>
> We thank the reviewer for the valuable feedback. We wish to note that our method has demonstrated improvements in all metrics on the KvasirCapsule-SEG dataset and showed an improvement of more than 2% in Dice score over the state-of-the-art AutoSAM on both SUN-SEG Easy and Hard datasets.
>
> We would like to highlight the major improvements our method provides to NanoNet which is a much smaller model than AutoSAM. By leveraging h (which is also small), NanoNet was able to drastically narrow the performance gap from AutoSAM.
>
> Lastly, We want to emphasize that our method does not require training the base model thus it is time and cost-efficient while providing a high degree of modularity.
>
> We are happy to provide additional information regarding the paper.
>
>
> >Input of the h-network: is it the raw values of the image at (x,y,t) or the x,y,t coordinates in the video? i.e. is it (x,y,t) or V(x,y,t)? This information is not clear in the text and in Figure 1 it seems it is the coordinates. If so, how are these coordinates normalised?
>
> The input to the network is (x,y,t). The implicit model provides an output in the form of h(x,y,t). (x,y,t) are normalized in the range of (-1,1). We will add this clarification in the next version of the manuscript.
>
>
> >How do the Fourier features relate to these coordinates? Are they local? I would suggest the authors to further explain these details in the text.
>
> Each coordinate (x,y,t) is converted to a vector of N=16 Fourier Features represented by an exponential distribution between base frequency and the $2^N$ harmony. This scale allows us to emphasize the representation of the lower frequencies where most of the data is present. Each frequency is represented by a pair {$\{\sin(n\pi x), \cos(n\pi x)\}$} (for example here the x coordinate is presented). This information alongside further clarifications will be added to the revision of the manuscript.
>
>
> >Optical flow: the approach used to compute the optical flow was not trained for this domain, so how much does it contribute to improving the performance? Also, are the authors using the directed optical flow or just the magnitude of Delta X and Delta Y?
>
> Correct, RAFT was not fine-tuned for this specific task. To investigate the contribution of the optical-flow mechanism we have conducted a series of tests, as demonstrated in the ablation subsection (last part of section 4). This study shows the contribution of the Opitcal-Flow mechanism to our method. Although running the method without it $(\lambda_{\text{of}} = 0)$ yields some improvement over the baseline, the major benefit is demonstrated with the optical flow loss being incorporated into the method.
> We are using directed Optical-Flow to estimate the motion vector of each pixel from one frame to the following frame. This means  $\Delta X^t,\Delta Y^t \in \mathbb{R}^{W\times H}$ ,as mentioned in section 3, can have both positive and negative values.
>
>
> >h-training: it is meant to happen in inference with test images, then is it trained on each video or for the entire test set? Is there any data class balance for the selection of (x,y,t)?
>
> During inference, h is being trained on each test video separately, yielding a set of masks for this specific video for benchmarking. At each step in the training, (x,y,t) are randomly selected from a uniform distribution in the range (-1,1). We will add this clarification to the next version of the manuscript.
>
> >Benchmark:
> >- Were the approaches chosen for comparison retrained for this specific data domain? If not, are any of those supposed to be generalisable such as the foundation model AutoSAM? If so, to provide a fair comparison, it would be recommendable to indicate which ones were retrained and which ones are meant to be generalisable across domains.
> >- If possible, I would recommend the authors cite each of the approaches used for comparison.
> >- The table only contains mean values and many of them are quite similar, which impedes any conclusion regarding the performance of the method. A solution is to provide plots showing the distribution of these accuracies or the standard deviations of each of these metrics.
> >- Were the accuracy metrics for the image-based and video-based approaches computed in the same way? That is, where the accuracy metrics compute for each video and then averaged, or were all the metrics averaged for all the frames?
>
> - All Methods are retrained for the specific task.
> - All Methods are cited at the start of the results section.
> - The values shown in Table 1. are taken as is from the literature. Since only mean values were published, further statistical analysis is impossible.
> - Both image-based and video-based results were computed in the same way. Results were computed for each video frame, then averaged per video, and later averaged across all videos. This follows the evaluation script provided by VPS: https://github.com/GewelsJI/VPS/tree/main/eval (Ji et al. 2021)

---

### Official Review · Reviewer_PaRD · 2024-02-28

**Confidence:** 4
**Preliminary Rating:** 5
**Recommendation:** Oral
**Final Rating:** 4

**Summary:**

This work introduces a postprocessing method for frame-based video segmentation results that improves temporal consistency. The method is based on implicit networks, so a new network h is trained based on the frame-wise results of a segmentation network g to optimize a label loss (based on g's prediction) as well as an optical flow-based loss (based on the label consistency between consecutive frames, warped using vector fields from RAFT). The results are compared against many frame- and video-based methods.

**Strengths:**

The paper is very well-written and easy to read. The evaluation is relatively comprehensive – based on three datasets (two of which are related) and six measures each (not the same one, but apparently the same commonly used for the datasets). Two different architectural choices (actually four different parameterized models) are used for the underlying model g, ranging from small and cheap to expensive and SotA. The results are compelling. The method's applicability to other video data is given because it does not depend on the task that g performs.

**Weaknesses:**

The method requires training h on each new video at inference time, much like using CRFs for consistency postprocessing.

It is not stated how long the training of h takes and how much GPU RAM is necessary (of the 48GB available on the NVIDIA A6000 used).

The results of the method kind of depends on good optical flow results (but the employed RAFT algorithm seems to be a good choice).

**Detailed Comments:**

Re: Confidence – I am very confident about the statements I made, but I am not very familiar with the specific datasets and polyp segmentation task and the respective literature. I acknowledge that there are six measures used, but I am not sure how low-level or task-adequate they are (voxel-wise / frame-wise / polyp-wise?), but since they're apparently used in related works as well and this paper focuses on the implicit-network based postprocessing, I think it is not of primary importance. It would be interesting, though, to know if the reduced sensitivity means that a polyp was actually missed completely or just on a few frames.

The abstract and introduction claim that "pixel-wise accuracy" is important for this task, which makes me wonder: Isn't the main question to *detect* the polyps and the *exact* contour of secondary importance?

"BCE", "flow", "label" etc. in LaTeX math mode should be wrapped in text macros, since the default rendering is not very nice (broken kerning because it is interpreted as product of single-letter variables).

It is quite interesting that applying h with $\lambda_of=0$ still gives better results than g alone, but understandable when thinking about it.

**Justification Of Final Rating:**

I gave a very positive preliminary grade and still believe this to be an interesting contribution, but in light of the missing replies to our reviews (in particular to the more critical ones), I will go one step down to "weak accept".

**Justification Of The Preliminary Rating:**

As I wrote above, the paper is very well-written and presents a novel idea that seems to work (according to a plausible, relatively extensive evaluation).  Even if direct applicability may suffer from practical considerations, it may inspire future work also on other tasks.

**Questions To Address In The Rebuttal:**

Does "taking the last 20% of the dataset as test data" refer to the last 20% of frames in each video?  Or the last 20% of frames, and if so does that split a video?

Which split is used for SUN-SEG?  (I guess an official one, but that should be mentioned in the paper then.)

Why is the value of $\lambda_of=1$ (more or less) optimal?

Why do you use 1000 training iterations for SUN-SEG, but 500 for KvasirCapsule-SEG?

**Special Issue:**

No

---

> ### Author Response · Authors · 2024-03-10
>
> We thank the reviewer for providing constructive feedback. We are happy to provide additional information regarding the paper.
>
> >Does "taking the last 20% of the dataset as test data" refer to the last 20% of frames in each video? Or the last 20% of frames, and if so does that split a video? Which split is used for SUN-SEG? (I guess an official one, but that should be mentioned in the paper then.)
>
> We take the last 20% of the frames in the video. By doing so we split the video but keep the continuity in each subset of frames. Indeed, the split used in SUN-SEG is the official one. We will add this information in the revised version of the manuscript.
>
>
> >Why is the value of ${\lambda_{of}=1}$ (more or less) optimal?
>
> To our understanding, the reason the optimum is achieved around ${\lambda_{of}=1}$ might be by chance. But we can confirm that in this setting, ${L_{flow}}$ is about an order of magnitude smaller than ${L_{label}}$ which means that only a small refinement of the original mask is needed.
>
> >Why do you use 1000 training iterations for SUN-SEG, but 500 for KvasirCapsule-SEG?
>
> The videos provided by SUN-SEG are longer than KvasirCapsule-SEG. In each step of the training process, we sample at random a fixed amount of indices (x,y,t) which is determined by our compute constraint (VRAM).Therefore for longer videos we need more steps to get each pixel to be sampled approximately the same amount of times as in the shorter videos.

---

### Official Review · Reviewer_woeH · 2024-03-09

**Confidence:** 4
**Preliminary Rating:** 4
**Recommendation:** Poster
**Final Rating:** 2.5

**Summary:**

In this study, the authors present a method for the segmentation of Polyp on RGB videos. The authors propose the use of a segmentation model, combined with an implicit network to add time coherence between segmented frames. This method has been trained using a new Optical Flow-based loss.

**Strengths:**

The paper is well written. Authors made sure to compare their new approach to what has been published in the literature.  To the best of my knowledge, this is the first time that I have seen optical flow being used as a loss function and it is one of the main strengths of the study. The method has been tested on multiple datasets, using a wide range of evaluation metrics.

**Weaknesses:**

As I read the paper, I found It hard to conclude with absolute certainty that the proposed method outperforms an Image-Based AutoSAM as no confidence intervals of the metrics are reported and values remain very close with AutoSAM obtaining higher score averages on the first 4 evaluation metrics (Table 1). The addition of the implicit network seems to improve average scores on the KvasirCapsule-SEG, but with improvements in the order of 10e-3 in some metrics, please make sure to verify whether this difference is significant.
On a more conceptual level I wonder if such a method is applicable in a real-time segmentation task, as it takes as input the whole video it seems, as polyp video segmentation would prove mostly useful during medical interventions.

**Detailed Comments:**

- Please remove the listing of tested models from the results and move it to the experiments section.
- Please detail the procedure used to transform the vector (x, y, t) to Fourier features, although simple, from what distribution did you sample them?

**Justification Of Final Rating:**

I do not seem to find the author's answers on Open review as i do with other revisions. Unless it is an IT issue I am unable to better my grading of the paper nor accept it. Please address comments as soon as possible.

**Justification Of The Preliminary Rating:**

The use of optical flow loss is innovative and relevant in the case of video segmentation. Using implicit networks with Fourier features to capture high-frequency information and overcome spectral bias and therefore relevant. Both points are interesting topics from which other imaging specialists might benefit.

**Questions To Address In The Rebuttal:**

Cf weaknesses and detailed comments

**Special Issue:**

No

---

> ### Author Response · Authors · 2024-03-10
>
> We thank the reviewer for the valuable feedback. We kindly want to address all of the questions:
>
> >Please remove the listing of tested models from the results and move it to the experiments section.
>
> Thank you for the feedback. The listing will be moved in the revised version.
>
> >Please detail the procedure used to transform the vector (x, y, t) to Fourier features, although simple, from what distribution did you sample them?
>
> Each coordinate (x,y,t) is converted to a vector of N=16 Fourier Features represented by an exponential distribution between base frequency and the $2^N$ harmony. This scale allows us to emphasize the representation of the lower frequencies where most of the data is present. Each frequency is represented by a pair {$\{\sin(n\pi x), \cos(n\pi x)\}$} (in this example the x coordinate is presented). This information will be added to the revision of the manuscript.
>
>
> We wish to thank the reviewer again for their constructive feedback. With regards to the pointed weaknesses, we wish to highlight the Dice improvement in the SUN-SEG benchmark which is more than 2% over the state-of-the-art method AutoSAM. Also it is worth mentioning the major improvements our method provides to NanoNet which is a much smaller model than AutoSAM. By leveraging h (which is also small), NanoNet was able to drastically narrow the performance gap from AutoSAM.

---

### Author Response · Authors · 2024-04-07

General Response:

We thank the reviewers for their valuable feedback and suggestions.
We deeply regret any inconvenience caused by the technical issues that prevented some of our responses from being visible. We gratitude to the reviewers for initially providing highly positive scores and feedback. We deeply regret any downgrade caused by the technical issues that affected the visibility of some of our responses.

We assure the reviewers that their initial positive evaluation has not gone unnoticed, and we are committed to addressing their concerns and incorporating their valuable feedback into the revised version of our submission. We apologize for any inconvenience caused and express our sincere appreciation for their understanding and continued support.

Here's a summary of the changes we will make based on the reviewers' comments:

* In response to reviewer woeH and reviewer j1t2, we will provide an explanation on the utilization of Fourier Features.
* In consideration of reviewer j1t2's feedback, we will elaborate on the test-time training process of h.
* As per reviewer PaRD's recommendation, we will include information regarding the SUN-SEG dataset split.
* Responding to reviewer j1t2's input, we will clarify the input of h and the normalization of the xyt coordinates.
* Following reviewer woeH's suggestion, we will relocate the listing of tested models from the results to the experiments section.

Once again, we extend our gratitude to the reviewers for their valuable insights, and we are committed to incorporating their feedback to enhance the quality of our submission.

---

### Meta-Review · Area_Chair_8m8A · 2024-04-04

**Recommendation:** Accept (Poster)
**Confidence:** 3

**Metareview:**

The authors answered the reviewers' questions, sometimes too briefly and without any evidence that this will be incorporated into future articles. It would have been possible to update the manuscript and highlight the changes. However, I trust the authors to do this before final submission. I urge them to do so as soon as possible. I therefore suggest accepting the paper for publication in MIDL.

---

### Decision · Program_Chairs · 2024-04-06

Accept (Poster)